# Cytokine mRNA Expression Profile in Target Organs of IFNAR (-/-) Mice Infected with African Horse Sickness Virus

**DOI:** 10.3390/ijms25042065

**Published:** 2024-02-08

**Authors:** Eva Calvo-Pinilla, Luis Jiménez-Cabello, Sergio Utrilla-Trigo, Miguel Illescas-Amo, Javier Ortego

**Affiliations:** Centro de Investigación en Sanidad Animal (CISA), Instituto Nacional de Investigación y Tecnología Agraria y Alimentaria, Consejo Superior de Investigaciones Científicas (INIA-CSIC), 28130 Valdeolmos, Spain; lfj.cabello@inia.csic.es (L.J.-C.); sergio.utrilla@inia.csic.es (S.U.-T.); miguel.illescas.amo@gmail.com (M.I.-A.); ortego@inia.csic.es (J.O.)

**Keywords:** AHSV, pro-inflammatory cytokines, inflammation, cytokine storm, IFNAR (-/-) mouse, equine

## Abstract

African horse sickness (AHS) is a highly severe disease caused by a viral etiological agent, African horse sickness virus (AHSV). It is endemic in sub-Saharan Africa, while sporadic outbreaks have occurred in North Africa, Asia, and Europe, with the most recent cases in Thailand. AHSV transmission between equines occurs primarily by biting midges of the genus *Culicoides*, especially *C. imicola*, with a wide distribution globally. As research in horses is highly restricted due to a variety of factors, small laboratory animal models that reproduce clinical signs and pathology observed in natural infection of AHSV are highly needed. Here, we investigated the expression profile of several pro-inflammatory cytokines in target organs and serum of IFNAR (-/-) mice, to continue characterizing this established animal model and to go deep into the innate immune responses that are still needed.

## 1. Introduction

African horse sickness virus (AHSV) includes nine different serotypes classified within the genus *Orbivirus*, family *Sedoreoviridae* [1]. The orbiviral genome is composed of ten double-stranded RNA segments that encode for seven structural and five non-structural proteins. Although most RNA segments are monogenic, some of them have two or three functional open reading frames [2].

AHSV is a devastating pathogen affecting equines, producing high mortality rates in horses of non-endemic areas. The disease is currently enzootic in sub-Saharan Africa [3]. Its transmission outside enzootic regions of sub-Saharan Africa has resulted in many serious epizootics of disease in Spain, Portugal, India, Pakistan, the Arabian peninsula, and most recently in Thailand [3,4,5]. *Culicoides* biting midges (Diptera: Ceratopogonidae) efficiently transmit AHSV among equids. In particular, *C. imicola* and *C. bolitinos* appear to be the major biological vectors, with *C. imicola* sp. being the most widely distributed of all species of this genus [6].

Importantly, African horse sickness (AHS) seems to be one of the most lethal vector-borne viral diseases affecting horses. Disease presentation can be peracute, subacute, mixed, or a milder form of the disease known as horse sickness fever, being largely fatal in up to 90% of cases [7]. Thus, the development of countermeasures able to reduce AHS mortality should be a relevant topic for the research community. Effective vaccination strategies and anti-viral compounds would help to control the disease in risk areas. In this sense, a reliable laboratory animal model is essential for pilot experiments to evaluate vaccine candidates and antiviral treatments. In recent years, type I IFNAR receptor (-/-) knockout mice have been characterized as an animal model for AHSV infection and used for research purposes since the model recapitulates the severe disease observed in horses [8,9,10]. In natural hosts infected with AHSV, the incubation period is approximately 3 days to 2 weeks [7] and high viral multiplication is found in lungs, spleen and other lymphoid tissues [11]. Previous studies in the laboratory demonstrated that transgenic IFNAR (-/-) mice became highly susceptible to AHSV infection, showing viremia and high viral loads mainly in the spleen and lungs. Clinical signs appear in mice approximately 3–6 days after exposure to the virus [8,12]. This immunocompromised mouse model has been useful for the evaluation of prophylactic and therapeutic countermeasures as well as for pathology and immunobiology research [13,14,15,16,17,18,19,20,21,22,23].

Endothelial cells and monocyte-macrophages are described as the main target cells for AHSV replication [24]. The virus is also closely associated with erythrocytes that can lead to a prolonged viremia. Viral replication results in endothelial cell damage and monocytes/macrophage activation starting the cytokine production, leading to increased vascular permeability and leakage of fluid into the sub cutis and lungs [7]. Studies regarding the expression of pro-inflammatory cytokines that may contribute to AHSV pathogenesis are scarce. Recently, transcriptome analysis of RNA sequences to characterize innate immune responses activated during AHSV-4 have been explored in horse PBMCs after vaccination [25]. Nevertheless, characterization of pro-inflammatory cytokine expression in the IFNAR (-/-) mouse model in response to severe infection of AHSV has not been investigated yet. In this work, we looked into the expression profile of several cytokines in target organs of IFNAR (-/-) mice during the course of infection of two AHSV serotypes, AHSV-3 and AHSV-4. We have also investigated circulating cytokines in serum and characterized infection in organs and viremia at different times post-infection. This research will be helpful to enable the understanding of the characteristics of severe AHSV infection in target organs, determining which cytokines may be studied as markers of protection in future evaluations of vaccine efficacy or antiviral countermeasures.

## 2. Results

### 2.1. Viremia and Clinical Signs after Infection of IFNAR (-/-) Mice with AHSV-3 and AHSV-4

Previous studies in the laboratory demonstrated that a dose of 10^6^ plaque forming units (PFU) per mouse of AHSV-4 (viral isolate Madrid87) was capable to cause a fatal disease in IFNAR (-/-) mice [8]. Disease progression led to perceivable clinical signs and detectable viremia starting at 3 or 4 days post-infection (d.p.i.). To calculate the minimal lethal dose of AHSV-3 (viral isolate RSArah3/03), three groups of mice (n = 4) were inoculated subcutaneously (s.c.) with 10, 100 or 1000 PFU. With any dose of infection, mice were susceptible to viral infection, showing clinical signs of disease such as ruffled hair, ocular discharges and reduced activity. At higher doses, 100 and 1000 PFU, onset of clinical signs occurred at 3 d.p.i., and mice were sacrificed when they reached the humane endpoint, between 4 and 5 d.p.i. At the lower dose (10 PFU) animals developed clinical signs at 4 or 5 d.p.i., and were euthanized at 6 d.p.i. when they reached the humane endpoint. As all infected mice succumbed to the infection with these three different doses of AHSV-3, we did not infect more groups of mice with higher doses, and we determined that 10 PFU of AHSV-3 is a valuable dose for infection purposes.

Hereafter, we performed the following study to analyze the pro-inflammatory cytokine expression profile in target organs of AHSV infected mice. Thus, groups of mice (n = 5) were inoculated s.c. with a lethal dose of AHSV-3 (10 PFU) or AHSV-4 (10^6^ PFU). As AHSV-3 is more pathogenic than AHSV-4, samples were collected at 3 and 5 d.p.i. for this serotype, and at 3, 5 and 7 d.p.i. for AHSV-4. A mock-infected control group (n = 5) was included. At selected time points post-infection, mice were euthanized, and blood and organs were collected to evaluate viremia and viral loads in target organs. It is important to note that just four out of five mice inoculated with AHSV-4 that should be sacrificed at 7 d.p.i. were included, as one mouse was sacrificed at 6 d.p.i. due to the severity of disease.

Clinical signs were observed in animals from 4 d.p.i. in AHSV-3 inoculated animals and from 5 d.p.i. in AHSV-4 inoculated groups. These included eye swelling, ruffled hair and the reduction of mobility. We observed a drop in animal weight after inoculation with AHSV-3 or AHSV-4 compared to the control group (Figure 1A). Indeed, animals inoculated with AHSV-3 reached a 20% of body weight loss at 5 d.p.i. Similarly, animals infected with AHSV-4 suffered a significant reduction of weight at 7 d.p.i. compared to the control group (Figure 1A). Viremia was estimated by real time qRT-PCR as described in Materials and Methods, using a standard curve from serially diluted AHSV of known plaque titer. Estimated viral loads (PFU equivalents) by this method refer to viral RNA levels, not infectious virus. In our analyses, AHSV RNA was not detected in mock-infected animals (Figure 1B). In contrast, AHSV-3 and AHSV-4 infected mice displayed detectable RNA levels from 3 to 7 d.p.i. In groups infected with AHSV-3, increasing viremia levels were observed from 3 to 5 d.p.i., with an average viral load of 2.76 ± 1.4 and 3.94 ± 0.84 log PFU equivalents/mL at 3 and 5 d.p.i, respectively. In mice challenged with AHSV-4, viremia was detected from 3 d.p.i. (1.41 ± 1.16 log PFU equivalents/mL), escalating in subsequent days (3.20 ± 1.04 log PFU equivalents/mL), and reaching its highest average value (3.85 ± 0.49 log PFU equivalents/mL) at 7 d.p.i. (Figure 1B).

### 2.2. Measurement of Viral Burden in Organs after AHSV Infection

Target organs from mice were collected after 3, 5 or 7 d.p.i. to measure viral replication by real time RT-PCR. The lungs, spleen, thymus and liver were collected since previous studies in the laboratory showed these are the main organs where the virus replicates after an infection of IFNAR (-/-) mice with AHSV. As expected, no viral RNA was detected in any tested tissues from the non-infected mice as shown in Figure 2. In groups infected with AHSV-3, viral RNA was detected in the spleen and lungs of all infected mice at 3 and 5 d.p.i. (Figure 2A). Viral loads were estimated using a standard curve as described in Materials and Methods through comparison with previously titrated AHSV samples and are depicted in Figure 2 as log PFU equivalents per 0.1 g of organ. Viral load was up to 3.99 log PFU equivalents/0.1 g in the spleen and 4.91 in the lungs. A lower viral burden was found in liver and thymus samples. In the liver, AHSV RNA was found in all mice at 5 d.p.i. and in 4 out of 5 mice at 3 d.p.i. In the thymus, low viral RNA levels could be detected just in three animals, one at 3 d.p.i. and the other two at 5 d.p.i. (Figure 2A).

In groups infected with AHSV-4, positive RNA was observed in most spleens and lungs. Viral titer in the spleen was up to 4.08 log PFU equivalents/0.1 g and 4.98 in the lungs. Despite not all animals inoculated with AHSV-4 displaying detectable RNA levels in the spleen and lungs at 3 d.p.i., we observed an upsurge in subsequent days, detecting high levels of viral RNA in spleens and lungs from all infected mice at 5 and 7 d.p.i. (Figure 2B). In livers, lower viral loads were detected compared with spleen or lungs; however, all infected animals were positive at 5 dpi. Viral RNA detection in thymus samples was similar to those of AHSV-3, as viral RNA was found in just one mouse at 5 d.p.i. and in two mice at 7 d.p.i. (Figure 2B).

### 2.3. Signature of Pro-Inflammatory Cytokines mRNA in Targets Organs of AHSV

In this work, we aimed to characterize the profile of inflammatory cytokines in target organs after AHSV infection. Several pro-inflammatory cytokines were analyzed: IL-1β, IL-6, IL-12, *CXCL2*, TNF and IFN-γ by relative expression of mRNA in the lungs, spleen, thymus and liver, using GAPDH as a house keeping gene. The expression of transcripts encoding the pro-inflammatory cytokine are shown in Figure 3 as fold change over the control (AHSV infected/mock).

Overall, we did not observe a significant increased expression of IL-12 or TNF in infected animals compared to mock-infected mice in any of the analyzed organs. In the liver, from both AHSV-3 and AHSV-4 infected mice, a significantly higher expression of cytokines compared to mock-infected animals was not observed at any evaluated time point. The lung was the organ where a higher expression of pro-inflammatory cytokines was found. Compared with cytokine transcription in the lungs of mock-infected mice, we observed an induction of IL-1β, IL-6, CXCL2 and IFN-γ mRNA over the time points that were analyzed (Figure 3A). Both AHSV-3 and AHSV-4 induced a strong inflammatory cytokines production in lung, with significant increase of IL-6 at 3 and 5 d.p.i. for AHSV-3, and at 3 d.p.i. for AHSV-4. Levels of IL-6 RNA were up to 8.3-fold (∆∆Ct fold change) higher in infected mice compared to mock-infected animals. Moreover, a significant up-regulation of IL-1β was detected at 5 d.p.i. for both serotypes, with up to a 4.9-fold increase above mock-infected mice. CXCL2 was also significantly expressed in the lungs of AHSV-3 infected animals at 5 d.p.i. Despite being non-significant, CXCL2 transcription levels were notably increased also after the inoculation of AHSV-4 at 3 and 5 d.p.i. The expression of mRNA IFN-γ was markedly increased in AHSV-3 and AHSV-4 infected mice at 5 d.p.i., though not reaching statistically significant values. Overall, the statistical significance was higher for animals infected with AHSV-3 compared to AHSV-4, which may rely on the faster replication of AHSV-3.

We also detected significantly higher expression levels of cytokine IL-6 in the thymus from both AHSV-3 and AHSV-4 infected mice at 5 d.p.i. compared to mock-infected animals (Figure 3B). Despite this, we did not find statistically significant differences compared to non-infected mice. The IL-1β expression in the thymus was increased in some AHSV-3 and AHSV-4 infected animals at 5 d.p.i.

In the spleen, we found a significant up-regulation of IL-6 and IL-1β transcription levels in AHSV-3 infected animals at 5 d.p.i. (Figure 3C). In mice inoculated with AHSV-4, we observed an up-regulation of the expression of IL-1β at 5 and 7 d.p.i., although it was non-significant. Similarly, some AHSV-3 and AHSV-4 infected animals displayed non-significant higher transcription levels of CXCL2 and IFN-γ at 5 d.p.i.

### 2.4. AHSV Infection Increased Levels of Circulating Pro-Inflammatory Cytokines

Serum samples of mice were also tested by a Luminex assay to study differences in circulating pro-inflammatory cytokine levels between mock-infected animals and AHSV infected mice. Cytokine levels were calculated using the appropriate standard curves of each protein. Figure 4 shows the cytokines that increased their levels in infected mice versus mock at 5 d.p.i. Levels of IL-6 and IFN-γ were significantly increased in animals infected with any of the two serotypes of AHSV evaluated, with values up to 8.5-fold in IL-6 values and up to 5.9-fold for IFN-γ. Nonetheless, the average levels of these cytokines were higher in serum from AHSV-3 inoculated animals compared to the AHSV-4 infected group. There was also a notable increase in the circulating levels of other pro-inflammatory cytokines, such as IL-12 and TNF

## 3. Discussion

Previous works have demonstrated that transgenic IFNAR (-/-) mice are a suitable animal model to study some aspects of the disease produced by AHSV [12,26]. After infection of AHSV in an equine host, first replication starts in regional lymph nodes close to the site of the Culicoides bite. The progeny of the virus disseminates throughout the organism via the blood, reaching main target organs such as the lungs, spleen and other lymphoid tissues, where the virus replicates at high levels [11]. Similarly, we observed that the replication of AHSV also occurs in the lungs, spleen, thymus and liver of IFNAR (-/-) mice. After infection with two different serotypes of the virus, higher loads of AHSV were found in lungs and spleen. In most acute cases of AHSV in horses, pulmonary edema and pleural effusion are present, leading to respiratory distress [27,28].

The main target cells of AHSV include monocytes and endothelial cells. Immuno-histological studies in horse tissues suggested that monocytes and macrophages, including pulmonary intravascular macrophages, may play a relevant role that contributes in the pathogenesis during AHSV infection via production of pro-inflammatory cytokines [24,29]. Cytokines are defined as soluble protein mediators that are relevant for the orchestration of inflammatory responses of the organism [30,31]. These include interferons (IFNs), tumor necrosis factors (TNF), chemokines, and interleukins (IL). Common pro-inflammatory cytokines (IL-6, IL-1 and TNF) regulate the cell death of inflammatory tissues, modify vascular endothelial permeability, recruit blood cells to inflamed tissues, and induce the production of acute-phase proteins [30].

For bluetongue virus (BTV), a closely related virus to AHSV, it is described as an induction of a massive release of pro-inflammatory cytokines that results in cellular dysfunction [32]. Endothelial cells infected with BTV induce the production of IL-1, IL-8, TNF and IL-6 [32,33]. In goats infected with BTV, cells staining positive for pro-inflammatory cytokines (TNF-α and IL-1α) were abundant in the spleen, lymph nodes and lungs [34]. Also for BTV, infected PBMCs from sheep expressed significantly higher levels of IFN-γ and IL-6 at transcriptional levels [35]. Regarding AHSV, there is a lack of information about pro-inflammatory cytokines produced in target organs after a fatal infection, both in horses and in mice. More is known about cytokines induced after vaccination. In vaccinated horses, adaptive immune response could be identified in vitro by the production of cytokines by PMBCs that were indicative of T helper (Th)1, Th2 and Th17 responses [36]. Also, in immunized horses with an attenuated vaccine, the transcriptome analysis of immune responses in PBMCs after in vitro infection with AHSV revealed the up-regulation of genes of different cytokines such as IL-6, TNF, CXCL2 and IL-1β [25]. Here, we attempted the first study of the induction of pro-inflammatory cytokines in target organs after AHSV infection that may contribute to viral pathogenesis and can be used as markers of protection.

In our study, extensive changes in cytokine profiles were observed in organs from IFNAR (-/-) mice challenged with AHSV. These changes seem to be associated with severity of infection, as in general higher levels of cytokines were seen in AHSV-3 inoculated mice compared to AHSV-4 infected animals. AHSV-3 replicates faster in mice at a lower dose compared to AHSV-4, being a more pathogenic strain in IFNAR (-/-) mice. Mainly, higher expression of all cytokine mRNA was detected at 5 d.p.i. than at 3 d.p.i. or 7 d.p.i. The lung was the organ where higher changes were found, followed by the spleen and thymus. The higher induction of pro-inflammatory cytokines in the lung may correlate with the relevant role of pulmonary intravascular macrophages described for AHSV pathogenesis [29].

Interleukin-6 was the cytokine that was more increased in AHSV infected lungs compared to mock animals, with significant up-regulation at 3 and/or 5 d.p.i. in AHSV-3 or AHSV-4 infected mice, respectively. This cytokine is important in inflammation and induces synthesis of acute phase proteins [37]. It was also upregulated in the thymus and spleen at 5 d.p.i. in AHSV-4 and/or AHSV-3 mice. There was also a significant increase of this cytokine in sera from mice infected with any of these two serotypes.

Interleukin-1 beta (IL-1β) is pro-inflammatory cell-associated cytokine, produced by monocytes, tissue macrophages, keratinocytes and other epithelial cells [38]. IL-1 β promotes the proliferation of memory T cells and cytokine production. In our studies, IL-1β was one of the most elevated cytokines in the lung and spleen from AHSV infected mice. There was a significant expression of IL-1β in the lung at 5 d.p.i. for both infected groups, and in the spleen also at 5 d.p.i. for AHSV-3 infected mice. In the thymus, we detected elevated levels of IL-1β in a few infected animals. IL-1β in serum could not be assessed in this study. However, it will be analyzed in future experiments.

CXCL2 (C-X-C Motif Chemokine Ligand 2) is a cell-signaling cytokine and a chemoattractant chemokine secreted by endothelial cells and monocytes with pro-inflammatory function. CXCL2 is important for neutrophil recruitment during viral infection, which contributes to pathogen clearance, however, this is also closely related to severe inflammatory tissue damage during viral infections, and the worsening of the immunopathology associated with cytokine storm [39,40]. In our study, CXCL2 was also elevated in target organs compared to mock-infected animals. In particular, increased CXCL2 mRNA levels were detected in the lungs of most AHSV infected animals at any time point analyzed. Some infected animals also had higher CXCL2 expression in spleen. IFN-γ was also another upregulated cytokine found in the AHSV infected mice. In particular, the lungs of infected animals showed an increased expression of this cytokine at 5 d.p.i. (4 out of 5 AHSV-3 inoculated mice and all AHSV-4 infected mice). Moreover, significant elevated levels of IFN-γ were found in sera from the infected animals compared to the control mice. TNF and IL-12 were not found to be elevated in any of the organ samples analyzed; however, they were elevated in sera from infected animals.

Cytokines are made by many cell populations, but the predominant producers are helper T cells (Th) and monocytes/macrophages [41,42]. Monocytes are nonspecific markers of inflammation and different subpopulations are described, with pro- and anti-inflammatory properties [42]. During inflammation or infection, monocytes are recruited by chemokines into the tissues where they differentiate into macrophages and dendritic cells. Monocytes can also migrate into peripheral tissues where they function as effector cells [43]. They play important roles such as the phagocytosis of pathogens and the modulation of the innate and adaptive immune system by secreting cytokines including IL-1β, IL-6, TNF among others [43]. As monocytes are important target cell of AHSV, the infection of monocytes may give rise to an excessive inflammatory response [25]. Monocytes [24] and pulmonary intravascular macrophages were described to play a major role in contributing to disease pathogenesis during AHSV infection [29].

In the present work, we have characterized pro-inflammatory cytokines that are significantly over expressed in organs from AHSV infected mice, and also elevated in serum. Although equines are currently the best animal model to study AHSV infection, due to ethical and financial considerations, the IFNAR (-/-) mouse model is a valuable tool to study AHSV pathogenesis as it reproduces pathological aspects of viral infection. Furthermore, we suggest that these pro-inflammatory cytokines could be useful as markers of inflammation to screen potential vaccine and antiviral efficacy in addition to possible therapeutic targets.

## 4. Materials and Methods

### 4.1. Virus and Cells

Baby hamster kidney cells (BHK-21; ATCC catalog No. CCL-10), and green monkey kidney cells (Vero; ATCC catalog No. CCL-81), were grown in Dulbecco’s modified eagle medium (DMEM) (Biowest, Nuaillé, France) supplemented with antibiotics and 10% fetal calf serum (FCS) (Gibco, Waltham, MA, USA). AHSV-4 (Madrid/87) and AHSV-3 (RSArah3/03) stocks were generated by the infection of BHK-21 cells at MOI of 0.1 for 48 h. Standard virus titrations were performed in Vero cells by plaque assays and calculated in plaque-forming units (PFU) per milliliter. Ten-fold dilutions of viral stocks were inoculated into Vero semi-confluent monolayers grown in 12-well plates. After 1 h incubation, agar overlay (0.4% agar in DMEM 10% FCS) was added and plates were incubated for 5 days at 37 °C in 5% CO_2_. Plaques were fixed with 10% formaldehyde and visualized with 2% crystal violet.

### 4.2. Animals and Ethics Statement

Type I interferon receptor defective mice, IFNAR (-/-), on a 129 Sv/Ev background were bred in the animal care facility of the Department of Animal Reproduction at INIA and housed under pathogen-free conditions at the biosafety level 3 (BSL3) animal facilities in the Animal Health Research Center (CISA-INIA/CSIC), Madrid (Spain).

Mouse experiments were conducted followed the European Union guidelines 2010/63/UE and the Spanish Animal Welfare Law 32/2007. Experiments were approved by the Ethical Review Committee at the CISA-INIA and by Comunidad de Madrid (Permit number: PROEX 075.2/22).

Animals were inoculated s.c. with 200 µL of AHSV-3 (10 PFU) or AHSV-4 (10^6^ PFU), and culled by groups of five animals at 3, 5 or 7 d.p.i. by cervical dislocation. A non-inoculated control group was included (n = 5). Mice were closely observed every day following challenge and any mice that showed signs of distress, paralysis or more than 20% weight loss were euthanized immediately. Blood samples were collected by submandibular vein from mice at 0, 3, 5 and 7 d.p.i.

### 4.3. Measurement of Viral Burden in Target Organs and Blood by Real Time RT PCR

Bloods (100 µL) from mice were collected in EDTA tubes and used for RNA extraction with TRI Reagent (Sigma-Aldrich, St. Louis, MO, USA) following manufacturer protocol. The organs analyzed (spleen, lung, liver and thymus) were collected, weighed, and homogenized in 0.8 mL of RPMI media containing 100 U/mL penicillin, 0.1 mg/mL-streptomycin and homogenized with a TissueLyser II (QIAGEN, Hilden, Germany) in 2 mL centrifuge tubes containing 1.5 mm zirconium beads. Virus titers were estimated in homogenized organs by quantitative real-time PCR (qRT-PCR) by a reference method using primers and probe described in the bibliography [44]. Primers and probe were directed to a highly conserved sequence within the segment 7 of the AHSV genome. Forward and reverse primers sequence are 5′-CCAGTAGGCCAGATCAACAG-3′ and 5′-CTAATGAAAGCGGTGACCGT-3′, respectively. The probe consisted of 5′-FAM-GCTAGCAGCCTACCACTA-MGB-3′ (Sigma-Aldrich, St. Louis, MO, USA). Amplification conditions consisted of a first reverse-transcription step at 55 °C for 30 min, followed by 15 min at 95 °C, and 45 cycles of 15 s at 94 °C, and 1 min at 60 °C. Ct values lower than 38 were considered positive. Cts values were transformed to viral titers (PFU) using a standard curve with known dilutions of an AHSV infected cell culture stock (10^6^, 10^5^, 10^4^, 10^3^, 10^2^, 10 PFU).

### 4.4. Analysis of Inflammatory Cytokines mRNA

For cytokine expression analyses, RNA transcripts corresponding to several inflammatory cytokines (IL-1β, IL-6, IL-12, CXCL2, TNF-α, and IFN-γ) were measured by comparative RT-PCR. Total RNA was purified from 500 μL of the homogenized tissues in TRI reagent and then was reverse transcribed using High Retrotranscriptase Starter Kit with Oligo dT (Biotools, Madrid, Spain) to synthesize the first strand cDNA following the manufacturer’s instructions.

The relative quantification of pro-inflammatory cytokines was performed by the ∆Ct (Ct of gene of interest-Ct housekeeping gene) method using GAPDH as a housekeeping gene and PrimeTime Std qPCR Assays (Integrated DNA Technologies, Leuven, Belgium): Mm.PT.39a.1 for GAPDH, Mm.PT.58.10005566 for IL-6, Mm.PT.58.12575861 for TNF-α, Mm.PT.58.41616450 for IL-1 β, Mm.PT.58.10005566 for IFN-γ, and Mm.PT.58.10456839 for CXCL2. The results were expressed as fold change over the control (∆Ct AHSV infected/∆Ct mock). Samples with negative results in PCR that fell below the level of detection of the assay were assigned Ct 45. Amplification conditions consisted of 50 °C for 2 min, followed by 10 min at 95 °C, and 45 cycles of 15 s at 95 °C, and 60 s at 60 °C. Fluorescence data were acquired at the end step. RT-qPCR was performed on a ECOTM Real-Time thermal cycler (Illumina^®^, San Diego, CA, USA).

### 4.5. Cytokine Analysis in Serum

Sera from AHSV infected and non-infected mice collected at 5 dpi were used in the assays. Cytokine levels were analyzed using a multiplex fluorescent bead immunoassay for quantitative detection of mouse cytokines (Millipore’s 6-Plex IL12p70, TNF, IL-4, IL-5, IFN-γ and IL-6 MILLIPLEX Mouse Cytokine kit, Burlington, MA, USA) following the manufacturer’s instructions. Samples were analyzed with a MAGPIX system (Luminex Corporation, Austin, TX, USA) and data were collected using xPonent version 3.1 software. The cytokine concentrations were calculated using standard curves. Values that fell below the level of detection of the assay were assigned the lowest detectable concentration.

### 4.6. Statistical Analysis

Data were analyzed using GraphPad Prism software version 9.0.1 (GraphPad Software, San Diego, CA, USA). Data from the viremia levels were analyzed using the Kruskal–Wallis test. The data from viral loads in organs, circulating cytokines in sera and cytokine expression levels in organs were analyzed using two-way ANOVA. The *p*-values lower than 0.05 were considered significant in all cases.

## Figures and Tables

**Figure 1 ijms-25-02065-f001:**
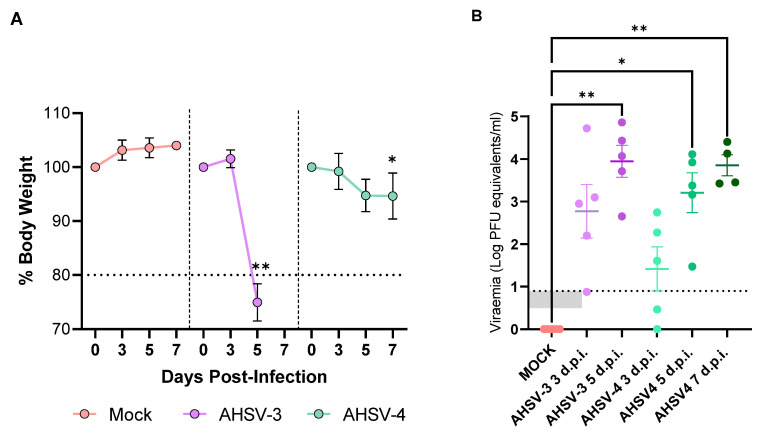
Body weight and viremia in mock infected or AHSV infected mice throughout the experiment. (**A**) Changes in animal weight. The shaded area represents the percentage of weight loss that reaches the humane end point. AHSV-3 mice include groups sacrificed at 3 and 5 d.p.i. AHSV-4 mice includes groups sacrificed at 3, 5 and 7 d.p.i. (**B**) Viremia in mice. Viral load in blood of mice infected with AHSV-3 and AHSV-4 at 3, 5 and 7 d.p.i. calculated by real time RT-qPCR. Results are expressed as log PFU equivalents/mL, converted from cycle threshold using a standard curve by comparison with previously titrated samples. The shaded area represents the cut-off of the assay (0.89 log PFU/mL). Asterisks represent statistical significance: * *p*-value < 0.05; ** *p*-value < 0.033, and standard error of the mean (SEM) is depicted by error bars.

**Figure 2 ijms-25-02065-f002:**
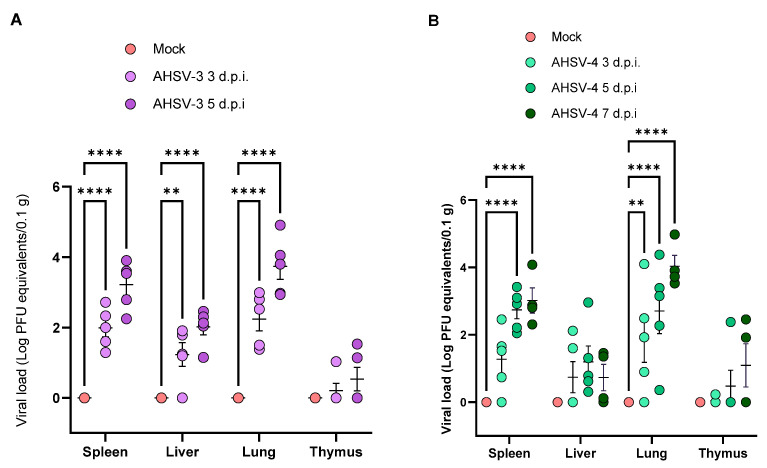
Viral loads detected in organs after infection of mice with AHSV-3 (**A**) or AHSV-4 (**B**). Viral RNA detected by real time RT-PCR after 45 cycles using protocol described in Section 4. Results are expressed Log PFU equivalents per 0.1 g of organ, converted from cycle threshold using a standard curve. Asterisks represent statistical significance: ** *p*-value < 0.033; **** *p*-value < 0.0001. SEM is depicted by error bars.

**Figure 3 ijms-25-02065-f003:**
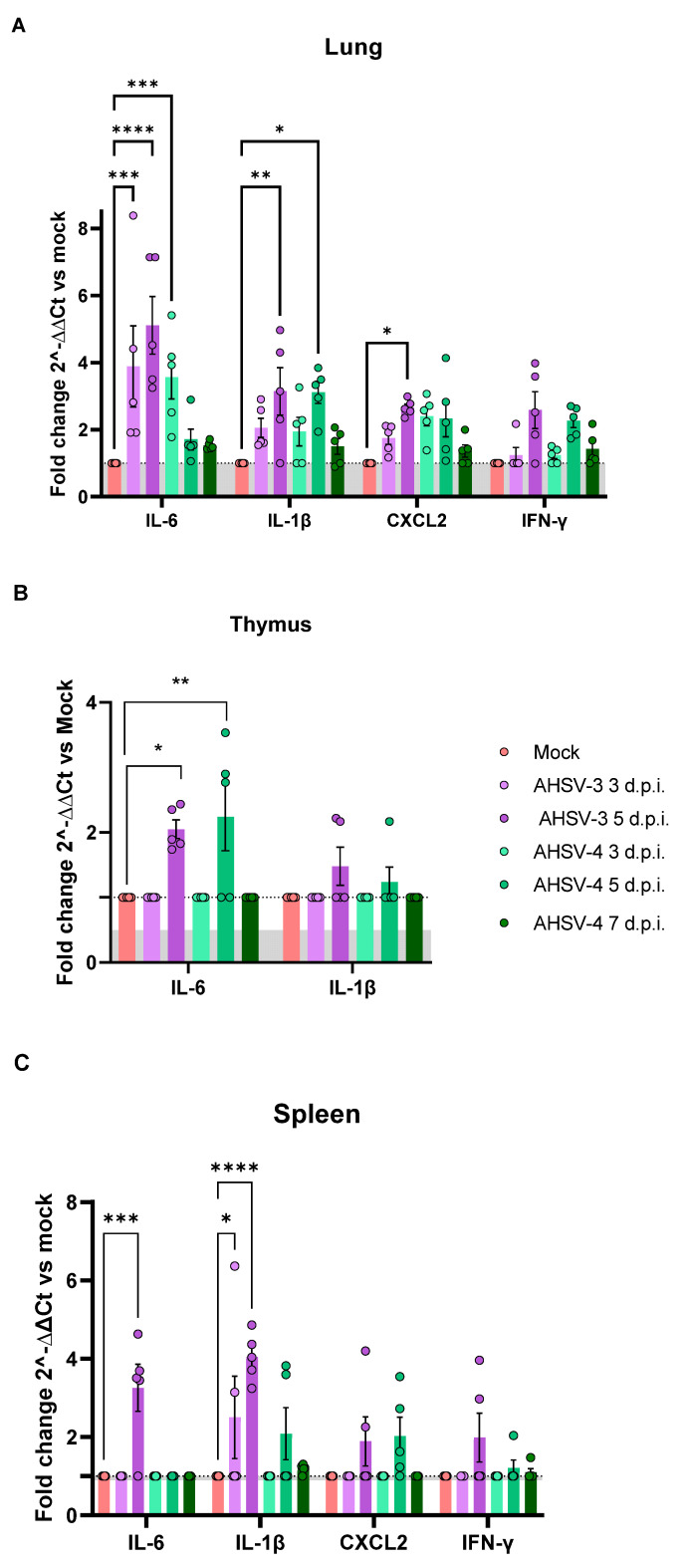
Profile of pro-inflammatory cytokines in lung (**A**), thymus (**B**) and spleen (**C**) after AHSV infection in mice. Comparative analyses were performed by qPCR analysis of cytokine gene expression using cDNA samples. Values of Ct were compared with a housekeeping gene by the 2^−∆∆Ct^ method. Fold changes values were compared to mock animals (Infected/mock). The dotted line represents the cut-off=1 (mock/mock). Asterisks represent statistical significance: * *p*-value < 0.05; ** *p*-value < 0.033; *** *p*-value < 0.002; **** *p*-value < 0.0001. SEM is depicted by error bars.

**Figure 4 ijms-25-02065-f004:**
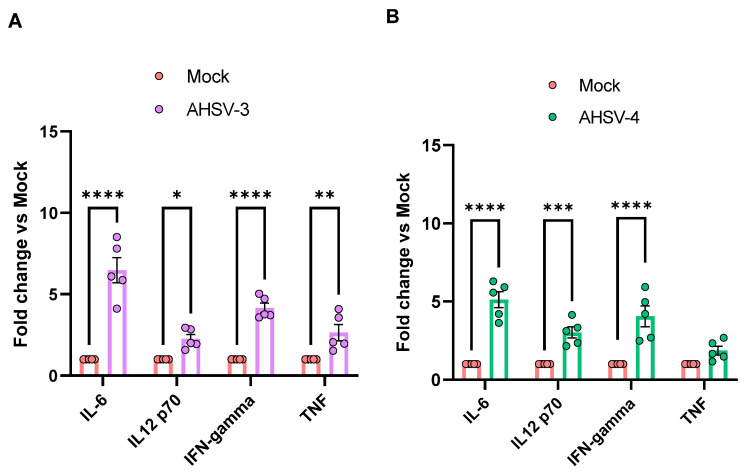
Changes in pro-inflammatory cytokines in serum after (**A**) AHSV-3 or (**B**) AHSV-4 infection in IFNAR (-/-) mice. Levels of protein were analyzed at 5 d.p.i. by a Luminex assay (pg/mL) and values were expressed as fold change compared to mock animals. Asterisks represent statistical significance: * *p*-value < 0.05; ** *p*-value < 0.033; *** *p*-value < 0.002; **** *p*-value < 0.0001. SEM is depicted by error bars.

## Data Availability

The data presented in this study are available on request from the corresponding author.

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
