# Peer review of "Cytokine mRNA Expression Profile in Target Organs of IFNAR (-/-) Mice Infected with African Horse Sickness Virus"

_ijms, 2024, doi:10.3390/ijms25042065_

Round 1

Reviewer 1 Report

Comments and Suggestions for Authors

I commend the authors for their efforts in investigating the mRNA expression profile in several cytokines in IFNAR mice infected with the African horse sickness virus (AHSV). 

I would like to make a few remarks and some recommendations.

Introduction

It would be good to mention the incubation period of the disease in horses. Is it similar to that observed in mice after inoculation with AHSV, or is it different?

Results

Line 69

“Previous studies in the laboratory demonstrated that a dose of 106 plaque forming units (PFU) per mouse of AHSV 4 (viral isolate Madrid8 7 ) was capable to cause a fatal disease in IFNAR ( (--//--) mice. Disease progression led to perceivable clinical signs and detectable viremia starting at 3 or 4 days post infection ( d.p.i. i.)”

Which studies? Unpublished results? Please mention.

Line 72

“To calculate the minimal lethal dose of AHSV 3 viral isolate RSArah3 /03 ), three groups of mice ( 4 ) were inoculated subcutaneously ( s.c. with 10, 100 or 1 00 0 PFU)”

Why was the dose of 106 not taken into consideration taking into account that for the other virus isolate (Madrid87) you used this dose? I understand that ASHV-3 is much more pathogenic than ASHV-4 in horses, but this is well-known in mice as well? I suggest you to clarify this aspect.

Line 319

How was the blood collected? Please mention

Author Response

We sincerely thank the reviewers for the positive comments on our manuscript. The reviewers raised a few issues that needed to be addressed or clarified. We have modified the manuscript and figures according to the comments. Below, please find a point-by-point response:

Reviewer 1:

I commend the authors for their efforts in investigating the mRNA expression profile in several cytokines in IFNAR mice infected with the African horse sickness virus (AHSV). 

I would like to make a few remarks and some recommendations.

Introduction

It would be good to mention the incubation period of the disease in horses. Is it similar to that observed in mice after inoculation with AHSV, or is it different?

Answer: 

This information has been included in the manuscript.

Line 45: The incubation period for African horse sickness in equids is approximately 3 days to 2 weeks (Mellor PS, Hamblin C. Vet Res. 2004).

Line 49: Clinical signs appear in mice approximately 3-6 days after exposure to the virus (Castillo-Olivares et al 2011). 

Results

Line 69

“Previous studies in the laboratory demonstrated that a dose of 106 plaque forming units (PFU) per mouse of AHSV 4 (viral isolate Madrid8 7) was capable to cause a fatal disease in IFNAR ( (--//--) mice. Disease progression led to perceivable clinical signs and detectable viremia starting at 3 or 4 days post infection ( d.p.i.)”

Which studies? Unpublished results? Please mention.

Answer: These previous data were described in the article: Castillo-Olivares et al, 2011. This reference has been added to the text.

Line 72

“To calculate the minimal lethal dose of AHSV 3 viral isolate RSArah3 /03), three groups of mice ( 4 ) were inoculated subcutaneously ( s.c. with 10, 100 or 1 000 PFU)”

Why was the dose of 106 not taken into consideration taking into account that for the other virus isolate (Madrid87) you used this dose? I understand that ASHV-3 is much more pathogenic than ASHV-4 in horses, but this is well-known in mice as well? I suggest you to clarify this aspect.

Answer: As all infected IFNAR mice succumbed to the infection with these three different doses of  AHSV-3, we did not infect more groups of mice with higher doses, thus we reduced the number of animals in these first experiments and we determined that 10 PFU of AHSV-3 is a valuable dose for infection purposes.

Line 319

How was the blood collected? Please mention

Answer: Blood samples were collected by submandibular vein from mice. We have included this information in the document (line 319).

Reviewer 2 Report

Comments and Suggestions for Authors

The manuscript entitled ’Cytokine mRNA expression profile in target organs of IFNAR (-/-) mice infected with African horse sickness virus’ by Eva Calvo-Pinilla et al. has investigated if IFNAR(-/-) mice could be an animal model to study AHSV infection by examining the AHSV infection level and the expression profile of several pro-inflammatory cytokines including IL-6, IL-1b, CXCL2, IFN-g . The aim of this study is clear but we need more clarification with the data presentation, besides, more experiments are needed to reach the study aims.

Major:

1.     Authors mentioned that different lethal doses of AHSV-3 and -4 have been used. Is there any difference with change of body weight, and how about viraemia level at different infectious doses? In Figure 1, which infectious dose has been used to present the results? Why standard deviation of viraemia is so big? Does it mean the results from different groups with different infectious doses have been included together for the data analyses? How to explain in mock control one sample has 10 PFU/mL, when was this in mock control, 0, 3, 5,7 dpi?

2.      In Figure 1B, how author did the conversion between viral titer (Log PFU/mL) and rt-pcr (copies/mL). More virological assay such as titration or plaque assay should be done for the blood samples to understand the change of viraemia level (infectivity).

3.     In result part 2.2, the expression of viral protein should be added.

4.     Figure 2 suggested AHSV infects immunological cells. From figure 3, IL-6 and IL-1b play roles in AHSV infection, further suggesting that macrophage cells are the target cells during AHSV infection. Have authors thought about to isolate PBMC and see the change of cytokine mRNA expression profile in a higher resolution as compared to current data presentation.

Minor:

Line 32, major instead of mayor

Figure 3 and 4, 2^-delta delta CT instead of delta CT

Comments on the Quality of English Language

Minor editing of English language required

Author Response

We sincerely thank the reviewers for the positive comments on our manuscript. The reviewers raised a few issues that needed to be addressed or clarified. We have modified the manuscript and figures according to the comments. Below, please find a point-by-point response:

  1. Authors mentioned that different lethal doses of AHSV-3 and -4 have been used. Is there any difference with change of body weight, and how about viraemia level at different infectious doses? In Figure 1, which infectious dose has been used to present the results? Why standard deviation of viraemia is so big? Does it mean the results from different groups with different infectious doses have been included together for the data analyses? How to explain in mock control one sample has 10 PFU/mL, when was this in mock control, 0, 3, 5,7 dpi?

Answer: AHSV-3 infectious dose experiment (not showed in figures) was necessary to know the accurate minimal lethal dose to infect mice and observe clinical signs and viremia. We summarized the most relevant results in lines 72-84. However, we do not include here information of viremia or weight of these three groups because we considered that this do not add important information to the aim of the work relating to the expression profile of cytokines.

We have clarified (line 85) that once the infectious lethal dose was determined as 10 PFU, we carried out the experiment of expression of cytokines that was the major aim of the work. This experiment included two groups infected with AHSV-3 (sacrificed at 3 and 5 dpi, no group for 7 dpi because mice will not survive so long), 3 groups infected with AHSV 4 (sacrificed at 3, 5 and 7 dpi); and a mock infected group (sacrificed at day 7 of the experiment). All data of this main experiment are showed in Figures 1-4.

Deviation of viremia was initially showed as SD, now has been changed to SEM. In the figure 1B we added: “and standard error of the mean (SEM) is depicted by error bars.” The results from different groups with different infectious doses have been represented separately in viremia graph. The groups of the same serotype were represented together just in the body weigh graph. Then we have added to the legend of figure 1A: “AHSV-3 mice include groups sacrificed at 3 and 5 d.p.i. AHSV-4 mice includes groups sacrificed at 3, 5 and 7 d.p.i.”

In figure 1B, there was a mistake in the value of a mock mouse (log 0.89 pfu/ml, cut off of the assay) because this sample was negative in real time AHSV specific RTqPCR as the rest of the mock animals. We have changed this result in figure 1B.

We added in line 125: “Viral titers were estimated using a standard curve as described in materials and methods”

And in line 347: Cts values were transformed to viral titers (PFU) using a standard curve with known dilutions of an AHSV infected cell culture stock (106, 105, 104, 103, 102, 10 PFU).

  1. In Figure 1B, how author did the conversion between viral titer (Log PFU/mL) and rt-pcr (copies/mL). More virological assay such as titration or plaque assay should be done for the blood samples to understand the change of viraemia level (infectivity).

Answer: We apologize for the mistake, we meant Log PFU/0.1 g instead of Log RNA copies/0.1 g. This has been already changed in the manuscript (text and figure). Cts values were converted to PFU using a standard curve with known dilutions of an AHSV infected cell culture stock (106, 105, 104, 103, 102, 10 PFU).

In other experiments in the laboratory, we had evaluated viremia by plaque assay and RTqPCR values for AHSV in vivo infections, and we observed a good correlation of these parameters. Therefore, as we had limited blood volume for the assays, we decided to use it for RNA purification and serum extraction.

  1. In result part 2.2, the expression of viral protein should be added.

Answer: As the aim of the work was to study cytokine profiles, we did not collect tissues (in paraffin) for immunohistochemistry assays to detect viral protein expression. However, other works have already performed these kind of studies (Castillo-Olivares, 2011 and Jones et al, 2023)

  1. Figure 2 suggested AHSV infects immunological cells. From figure 3, IL-6 and IL-1b play roles in AHSV infection, further suggesting that macrophage cells are the target cells during AHSV infection. Have authors thought about to isolate PBMC and see the change of cytokine mRNA expression profile in a higher resolution as compared to current data presentation.

Answer: We agree with the reviewer that the expression cytokine profile in PBMCs can be a useful information. In vitro release of cytokines by PBMCs may serve as a measure of their activation in vivo. However, we consider that levels of cytokines in serum are also a valuable data. The changes in serum inflammatory cytokines can be useful to analyse protective efficacy in potential vaccination experiments.

In addition, PBMCs purification was not possible to carry out in these experiments because of the limited volume of blood collected. When blood samples were collected, we used 50 µl per RNA extraction and the remaining volume was used for serum extraction. The serum volume that is needed for luminex assays is 25 µl (or 50 µl in duplicate).

Minor:

Line 32, major instead of mayor

This has been changed.

Figure 3 and 4, 2^-delta delta CT instead of delta CT

This has been changed.

Minor editing of English language required

English language writing has been reviewed.

We thank the reviewers for the comments and hope that the modifications introduced have improved the quality of the manuscript to be now acceptable for publication.

Round 2

Reviewer 2 Report

Comments and Suggestions for Authors

Please clarify the volume of inoaculation in the animal model in lines 83, 326. I do not agree that 'viral RNA copies numbers/g or mL' can be converted as 'PFU/g or mL', and later one is to determine the infectious doses.

Author Response

Please clarify the volume of inoculation in the animal model in lines 83, 326. I do not agree that 'viral RNA copies numbers/g or mL' can be converted as 'PFU/g or mL', and later one is to determine the infectious doses.

Answer:

Thank you for helping us to improve the manuscript. We apologize for the mistakes and sincerely thank you for your constructive comments.

We added the volume of inoculation (200 µL) in line 326. We considered that it is not completely necessary to add the volume again in line 83, as we described it in materials and methods. However, we will agree to add the volume in line 83 if the reviewer considers that this is important.

Regarding estimation of viral loads in PFU, we agree with the reviewer that this is not infectious virus. We apologize since this was not the right description and indeed, we meant PFU equivalents, not just PFU. We hope that changes in the text will clarify this point. Similarly to the estimated quantification that we have done in the work, many articles in the bibliography refer to estimation of viral loads as PFU/equivalents using a standard curve PCR, such as:

Infectious SARS-CoV-2 in Feces of Patient with Severe COVID-19. Emerg Infect Dis. 2020 Aug;26(8):1920-1922.doi: 10.3201/eid2608.200681.

Real-time polymerase chain reaction as a rapid and efficient alternative to estimation of picornavirus titers by tissue culture infectious dose 50% or plaque forming units. Microbiol Immunol. 2009 Mar;53(3):149-54. doi: 10.1111/j.1348-0421.2009.00107.x.

Pharmacological Elevation of Cellular Dihydrosphingomyelin Provides a Novel Antiviral Strategy against West Nile Virus Infection. Antimicrob Agents Chemother. 2023 Apr 18;67(4):e0168722. doi: 10.1128/aac.01687-22.

We have added several comments in the text to clarify this point:

Line 101-103: Viremia was estimated by real time qRT-PCR as described in materials and methods, using a standard curve from serially diluted AHSV of known plaque titer. Estimated viral loads (PFU equivalents) by this method refer to  viral RNA levels, not infectious virus.

Line 129: Viral titers loads were estimated using a standard curve as described in materials and methods, by comparison with previously titrated AHSV samples

Log PFU has been modified by Log PFU equivalents in lines 107, 109, 110, 131, 132, 137, Figure 1 and 2.

We hope that these changes would be enough to clarify this point. Otherwise, just Ct values can be shown as an alternative, but substantial changes have to be done in figures and manuscript. We sincerely think that PFU equivalents are accepted estimation values, and hopefully the reviewer will agree with this.
